# Energy Saving and Emission Reduction Potential Evaluation of a Coal Mine Based on Fuzzy Hierarchical Analysis

Fenfang Xu [1], Ruili Hu [2], Minbo Zhang [1], Weizhong Zhang [1,*], Qinrong Kang [1] and Mengzhen Du [1]

[1] School of Resource and Security Engineering, Wuhan Institute of Technology, Wuhan 430074, China; xff18365312797@163.com (F.X.); zhangminbo2015@163.com (M.Z.); kangqinrong02@163.com (Q.K.); dumengzhen0508@163.com (M.D.)

[2] Department of Security, University of International Business and Economics, Beijing 100029, China; huruili2015@uibe.edu.cn

[*] Correspondence: zhangweizhong2023@163.com

**Abstract:** Due to the non-renewability of coal resources and the effect of excessive resource loss on the economic development of enterprises forming a constraint, breakthroughs in high energy consumption to achieve energy saving and emission reduction in coal mines represent an important way to promote the development of coal enterprises. This paper takes the energy-saving work of the Wuyang coal mine, which is very representative of China's Shanxi Province, as an example, and uses hierarchical analysis to establish an assessment system for energy conservation and emission reduction in coal mines. We adopted on-site research, expert scoring, and project guidance to divide the factors affecting energy conservation and emission reduction in the Wuyang coal mine into 24 categories, and we then used fuzzy mathematical calculations to assess potential, which is of better practicability. The study shows that the grade of energy saving and emission reduction in the Wuyang coal mine is of a medium level, and the potential of energy saving and emission reduction is large. The main influencing factors are improving the recovery rate of refined coal, renovating pressurized filters, eliminating high-energy-consuming equipment, optimizing the parking process, replacing the flotation machine, and ensuring the intelligent management of power metering. According to the relevant factors derived from the energy-saving and emission reduction work at the Wuyang coal mine for implementing the relevant measures, we provide a reference for the work of energy saving and emission reduction in mines in order to promote the sustainable development of the coal mining industry.

**Keywords:** energy conservation and emission reduction; analytic hierarchy process; fuzzy comprehensive evaluation method; potential assessment; management measure

## 1. Introduction

Mineral resources are the basis for human survival and sustainable development, and their irreplaceability determines the inevitability of mineral resource development [1]. The study of energy conservation and emission reduction in mines can not only improve the economic efficiency of enterprises, but also improve the overall benefits to society. Coal mining enterprises are the main energy-consuming industries in China, and the implementation of energy-saving and emission reduction actions for the coal mining industry is an important implementation point for national policy measures. The construction of an energy-saving and emission reduction system for coal mining enterprises is a very complex process that requires the analysis of its characteristics and connotations and the clarification of the significance of each indicator.

Due to the complexity of mine production systems, it is difficult to make very detailed analyses of them by mere qualitative or quantitative evaluation, and the analyses results usually have large deviations. The fuzzy comprehensive evaluation method is a method to convert qualitative evaluation into quantitative evaluation, which is good at solving

the problem of fuzziness and of being difficult to quantify. One of the prerequisites for calculations using this method is to determine the weight value of each evaluation index, but when the evaluation index is complex and difficult to quantify, it is difficult to present a weight value directly to the index, and the hierarchical analysis method satisfies this drawback very well. The hierarchical analysis method decomposes the complex problem into several sub-problems and provides the decision data by comparing them two by two; and finally, the ranking weights can be given [2]. Van Laarhoven et al. [3] first proposed the method of combining fuzzy mathematical theory and hierarchical analysis for comprehensive evaluation. Based on the concept of building electrical designs based on hierarchical analysis and fuzzy synthesis theory to assess the energy-saving potential of housing buildings in a city in northern China, Qin [4] divided the information into five-criteria layers by using hierarchical analysis for weight calculations, and combined this with the fuzzy synthesis evaluation method for grade classification, concluding that the city's electrical energy efficiency is at a good level. Xin [5] used fuzzy mathematics and hierarchical analysis to propose a multi-criteria decision model of coal mine safety management using a comprehensive quantitative method and developed a set of factors affecting mine safety management using the method on a case study of the China Fushan Coal Company Limited; the evaluation results showed that the enterprise has a relatively good management model, but still needs to be strengthened. Fertat et al. [6] combined a hierarchical analysis and fuzzy integrated evaluation method to challenge the established OHS (occupational health and safety) maturity model, which can turn the qualitative information collected by experts into quantitative information, thus helping executives and managers to evaluate their OHS maturity through different management levels and make the right decisions to improve economic efficiency. Based on the existing energy-saving evaluation methods and the regional characteristics and climatic conditions of the area, Diao [7] selected the main factors of energy saving in local farmhouses to establish an index layer and used hierarchical analysis to derive the weights. They then applied the fuzzy comprehensive evaluation method to evaluate the farmhouses in Hebei Dam, combining this with the actual design of a heating system and envelope structure to provide energy savings. Cui [8] used the hierarchical analysis method to determine the weighting values of the main influencing factors in the "coal seam floor trap column sudden water hazard" evaluation model, combining the information detected by the coal mine in the early stage to determine the fuzzy set of the coal mine's sudden water evaluation system, and then used the fuzzy comprehensive evaluation method to find the level of its hazard and propose relevant measures and suggestions to reduce the hazard to ensure the economic benefits of coal mining enterprises. Gokce et al. [9] used spherical fuzzy hierarchical analysis to determine the level of sustainable industrialization in each country and used spherical fuzzy hierarchical analysis to analyze and rank the countries according to sustainable industrialization, creating a new perception of the sustainable industrialization capacity of EU countries. Zhang et al. [10] used the fuzzy comprehensive evaluation method for the comprehensive evaluation of soil pressure balance shield adaptability because the traditional hierarchical analysis method is determined by a decision-maker who decides the weighting data of the index weights, which is highly subjective. The improved hierarchical analysis method is created based on group decision-making, using the data of multiple decision-makers to analyze and finally derive the factors affecting shield adaptability.

By studying the literature related to the evaluation system of enterprise management at home and abroad, it was found that the current research on energy conservation evaluation mostly begins by using two aspects: the evaluation method and the construction of the index system. On the construction of energy-saving evaluation methods and energy-saving evaluation index systems, there have been many research results: the hierarchical analysis of complex problems can be decomposed layer by layer to derive the weights of the factors; the fuzzy comprehensive evaluation method is characterized by level division, which can be a fuzzy and complex situation to derive a clear evaluation; at present, there is insufficient research on the application of the two methods on the work of energy-saving

and emission reduction in coal mines, and the combination of hierarchical analysis and fuzzy comprehensive evaluation method is in line with the professional qualities and necessities of the coal mining industry research. Therefore, this paper combines the coal mining industry's energy-saving and emission reduction work with the characteristics of the needs of the Wuyang coal mine in China's Shanxi Province as an example of its energy-saving and emission reduction potential assessment. The weight of factors affecting energy saving and emission reduction was determined and ranked by the analytic hierarchy process (AHP). The fuzzy comprehensive evaluation method was then used for quantitative analysis. The influence factors at all levels and the evaluation grade of the energy-saving and emission reduction system in Wuyang coal mine are obtained, and the evaluation result of its energy-saving and emission reduction potential is obtained. By studying the evaluation method of this example, a set of decision-making models applicable to energy saving and emission reduction in mines is established, which will play a positive role in improving the efficiency of energy saving and emission reduction in the coal mining industry. The results of the study can provide new inspiration for the formulation of energy-saving and emission reduction action strategies for mines under the "dual-carbon target", which is conducive to solving the important problem of resource wastage in mines, and providing reference and guidance for the further improvement of the energy-saving effect of coal mining enterprises, with a view to realizing the long-lasting and stable development of coal mining enterprises [11,12].

## 2. Analytic Hierarchy Process and Fuzzy Comprehensive Evaluation

### 2.1. Fuzzy Hierarchical Analysis Flow Chart

Analytic hierarchy process (AHP) is a hierarchical weighted decision-making method based on systems theory [13,14]. When using AHP, it is necessary to conduct a systems analysis, divide the problem into different subgroups according to the objectives, then combine the components in conformity with different levels of composition based on the correlation between the components and the affiliations of the influence, forming a multi-level analytical model. The importance of each component is determined by comparing each component with each other, and then they are ranked in importance according to the judgment of the decision-makers [15]. The fuzzy comprehensive evaluation method is a specific application method of fuzzy mathematics, applying the principle of fuzzy transformation and the principle of maximum affiliation to quantitatively deal with some factors that are not easy to quantify or are very fuzzy and unclear, and to conduct a comprehensive evaluation on these factors [16,17]. In accordance with the hierarchical analysis method and fuzzy comprehensive evaluation method, a flow chart of the implementation of the two evaluation methods combined can be made (see Figure 1). The CI in Figure 1 is the criterion used to test the consistency of the judgment matrix.

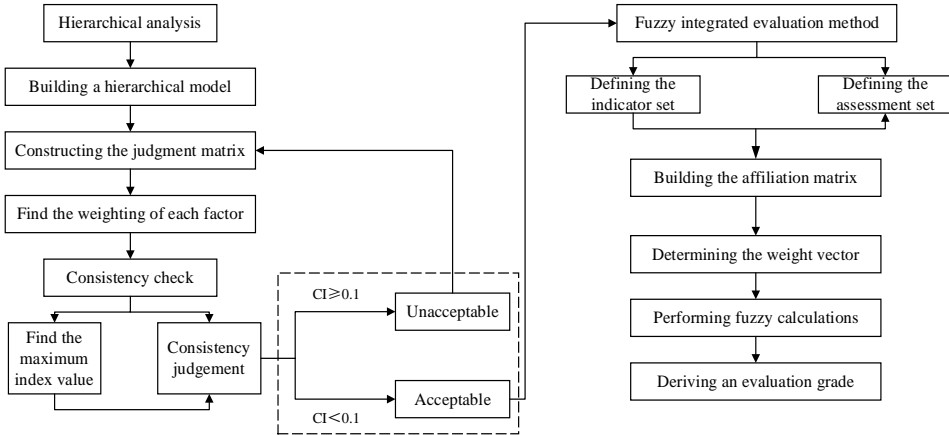

**Figure 1.** Fuzzy hierarchical analysis flow chart.

### 2.2. Analytic Hierarchy Process

The steps of the hierarchical analysis method are generally as follows:

(1)  The establishment of the hierarchical structure model

Decision-making objects and target are initially analyzed to determine the target layer, and then the middle layer and bottom layer are determined according to the interrelationship. The model consisting of the top layer, the middle layer, and the bottom layer is the hierarchical analysis structure model. Generally, the top layer of the model is the objective of the problem, the middle layer is the criterion of the factors, and the bottom layer is often the alternative of the decision.

(2)  Construction of judgment matrix

The judgment matrix of the problem is constructed by judging the relative importance of each factor according to each layer of analysis and expressing these judgments with appropriate scales and values. In order to form a judgment matrix, the scaling method is invoked, and the scales of the judgment matrix and their meanings are shown in Table 1.

**Table 1.** Scale and meaning.

| Scale | Meaning |
| --- | --- |
| 1 | Indicates that the two factors are of equal importance compared to each other |
| 3 | Indicates that one is slightly more important than the other when compared to two factors |
| 5 | Indicates that one is significantly more important than the other when compared to two factors |
| 7 | Indicates that one is strongly more important than the other when compared to two factors |
| 9 | Indicates the extreme importance of one over the other when compared to two factors |
| 2, 4, 6, 8 | The median of the above two adjacent judgments |
| Reciprocal | Comparing factor i with j yields judgment $b_{ij}$; comparing factor j with i yields judgment $b_{ji} = 1/b_{ij}$ |

(3)  Square root method to find weights

After multiplying the values in the judgment matrix by the rows, the product is squared n times to obtain the approximate value $w'_i$ of the weight vector of each evaluation factor in each line. Then, the n times values of each line are summed up, and $w'_i$ is divided by the sum of the n-times values of each line to obtain the weight vector $w_i$ of each factor.

$$w'_i = \left( \prod_{j=1}^{n} a_{ij} \right) 1/n (i = 1, 2, \ldots, n) \tag{1}$$

$$w_i = \frac{w'_i}{\sum_{k=1}^{n} \left( \prod_{j=1}^{n} a_{kj} \right)} > (1, 2, \ldots, n) \tag{2}$$

In the formula, $w'_i$ is the approximate value of the weight vector of evaluation factors in each row; $a_{ij}$ is the value in the judgment matrix; n is the number of elements in each layer; k is the layer number; and $w_i$ is the weight vector of each factor.

(4)  Consistency test

① Find the maximum eigenvalue using:

$$\lambda_{max} = \sum_{i=1}^{n} \frac{[Aw]_i}{nw_i} \tag{3}$$

In the formula, $\lambda_{max}$ is the maximum eigenvalue; A is the matrix generated from the judgment matrix.

② Consistency judgment

$$C{\cdot}R = \frac{C{\cdot}I}{R{\cdot}I} \tag{4}$$

$$C{\cdot}I = \frac{\lambda_{max} - n}{n - 1} (n > 1) \tag{5}$$

The R·I is a random consistency index, its values are shown in Table 2. The consistency is considered acceptable, otherwise, the judgment matrix needs to be adjusted until it is accepted.

**Table 2.** R·I consistency indicators.

| Matrix Orders | 3 | 4 | 5 | 6 | 7 | 8 | 9 | 10 | 11 | 12 |
|---|---|---|---|---|---|---|---|---|---|---|
| R·I | 0.52 | 0.89 | 1.12 | 1.26 | 1.36 | 1.41 | 1.46 | 1.49 | 1.52 | 1.54 |

*2.3. Fuzzy Comprehensive Evaluation Method*

The specific steps are as follows:

(1)    Determine the indicator set C

$$C = (C_1, C_2, \ldots, C_n)$$

(2)    Determine the collection of comments V

$$v = (v_1, v_2, \ldots, v_m)$$

The set of comments consists of the five grades which are excellent, good, medium, poor, and bad. Each grade can correspond to a fuzzy subset.

(3)    Build the affiliation matrix

After obtaining the rank fuzzy subset of each factor, each factor is quantified so as to determine the affiliation of the single factor to the rank fuzzy subset, and the following affiliation matrix is obtained.

$$R = \begin{pmatrix} r_{11} & r_{12} & \cdots & r_{1m} \\ r_{21} & r_{22} & \cdots & r_{2m} \\ \cdots & \cdots & \cdots & \cdots \\ r_{n1} & r_{n2} & \cdots & r_{nm} \end{pmatrix}$$

where R is the affiliation matrix; $r_{nm}$ is the affiliation degree of a single factor to a rank fuzzy subset.

(4)    Determine the weight vector W

According to the analytic hierarchy process, to determine the weight vector of each factor, $W = (w_1, w_2, \ldots, w_n)$, the relative importance among the components and the weighting coefficients should be determined and subsequently normalized before synthesis.

(5)    Perform fuzzy calculations

The membership grade is obtained by the weighted average method, and the fuzzy weight vector W and fuzzy relation matrix R are combined to obtain the fuzzy comprehensive evaluation vector B of each evaluation object.

$$B = W \cdot R = (w_1, w_2, \ldots, w_n) \cdot \begin{pmatrix} r_{11} & r_{12} & \cdots & r_{1m} \\ r_{21} & r_{22} & \cdots & r_{2m} \\ \cdots & \cdots & \cdots & \cdots \\ r_{n1} & r_{n2} & \cdots & r_{nm} \end{pmatrix} = (s_1, s_2, \ldots, s_n) \tag{6}$$

(6)    Derive the evaluation level

The comprehensive evaluation of the registration score is as follows:

$$f = \sum_{j=1}^{n} v_i \times B(i, j = 1, 2, \ldots, n) \tag{7}$$

In the formula, f is the overall evaluation score; $v_i$ is the decision level.

Calculate the corresponding grade score of each factor in the criteria layer, corresponding to the grade interval in the rubric set, and the corresponding risk grade is obtained.

## 3. Case Studies

### 3.1. Establishment of the Indicator System

In this paper, the indicators were selected according to the coal enterprise's energy-saving and emission reduction selection principles and in reference to its related laws and regulations, such as the "Coal Industry Green Mine Construction Specification", as well as the relevant standards and document requirements, such as the "Wuyang Coal Mine Energy Consumption Dual-control Construction Specification", and finally, by determining the reductions in coal washing losses, which reduces mine power consumption and mine heat consumption, but improves the basic measurement system of the mine. Additionally, the system management's five items refer to the first-level indicators of the energy-saving and emission reduction potential assessment. On the premise of following the feasibility, timeliness, and data availability of the indicators in accordance with the actual situation of energy-saving and emission reduction in the field inspection of mines, the relevant staff and experts in research and consultation should be contacted. Finally, with regard to China's energy-saving and emission reduction in mines and the relevant provisions and requirements, the energy consumption dual control action plan of the mine provided by the Wuyang coal mine enterprise should be combined, and the specific analysis is as follows.

(1)    Reduce coal washing loss

Coal washing process is the key link in coal refining. The basic process of coal washing in coal plant includes the following. ① The crushing and screening of coal. The raw coal is crushed by the mill, and the coal is divided into different particle size grades by the sieving machine after it is changed into a certain particle size. ② The soaking and stirring of coal. The coal powder is poured into a mixing tank for stirring and mixed with a certain amount of water and chemicals. ③ The reaction and flotation of coal. After the above steps, the impurities and ash in the coal will be chemically reacted to form some precipitates and scum, which shall be separated by gravity separation and flotation. ④ The dewatering and drying of coal. After gravity separation and flotation, the water content of coal is high. Generally, a centrifuge or filter press is used to remove water from coal. The coal is dried to a certain moisture content by using a dryer. In view of this, the factors affecting the source of energy consumption in the coal washing process mainly include the following. ① Technical factors: Technological improvement and consumption reduction can be accomplished through the following three aspects: the modification of a pressurized filter, the replacement of a flotation machine, and the use of centrifuges. ② Management reasons: Consumption reduction can be managed by improving the recovery rate of clean coal and optimizing the stopping process in these two aspects.

(2)    Reduce mine electricity consumption

The loss of electrical energy has always accounted for a large proportion in the production operations of mining enterprises. Due to the complexity of geological conditions, the reduction in mine electricity consumption should be safety-oriented. The process of reducing electricity consumption can be achieved through the renovation of electricity-using equipment and systems, which include the following six aspects: the elimination of high-energy-consuming equipment, a reduction in extraction time, the optimization of belt-conveying lines, the optimization of production systems, the renovation of drainage systems, and the enforcement of the decommissioning system.

(3)    Reduce heat consumption in mines

Equipment and facilities and staff bathing use a large amount of coal mine to generate heat, which leads to an excessive waste of resources. On the premise of ensuring the basic needs of employees, some old equipment and pipelines can be upgraded to reduce

unnecessary energy consumption. The heat consumption in mine can be reduced through technical and structural reduction in consumption, including the following five aspects: the utilization of the gas oxidation system, the modification of the water pump room, the modification of the pipe diameter pipeline, the modification of the heating machine, and the modification of the bathing hot water system.

(4)    Improve the basic metering system of the mine

By improving the basic metering system of the mine, the flow data used by the working face can be uploaded in a more timely manner, and the energy consumption can be better monitored. The construction of an intelligent system for power metering, the establishment of a hydrostatic water flow pressure monitoring platform, and the establishment of the pressurized air flow pressure monitoring platform are essential to monitor and understand the mine's energy consumption in an all-round intelligent way. Therefore, improving the basic metering system of the mine mainly includes the following four aspects: an intelligent management of power metering, system construction, the establishment of hydrostatic water flow pressure monitoring platform, and the establishment of an air flow pressure monitoring platform.

(5)    System Management

For an enterprise, the behavior of employees and the establishment of a sound mechanism lay a solid foundation for the smooth development and implementation of work. Through training, education, and an assessment mechanism to deepen the understanding of the mine's energy-saving and emission reduction, energy-saving and emission reduction combine rewards and punishment mechanisms to maximize employees' enthusiasm so that they will be conscious to comply with energy consumption management measures in the production process, and finally establish a sound energy consumption management mechanism. Therefore, the system management mainly includes four aspects: assessment mechanism, staff training and education, rewards and punishments mechanism, and the establishment of a sound energy consumption management mechanism.

### 3.2. Hierarchical Modeling

According to Section 3.1, it can be seen that the target layer consists of energy saving and emission reduction in Wuyang coal mine, and the criterion layer is divided into five first-level evaluation indexes. On the basis of further research on coal mine enterprises and relevant experts, 24 s-level indexes are finally determined as the index layer of the evaluation system, which constructs the hierarchical structure model of the evaluation system of energy-saving and emission reduction in Wuyang coal mine, as shown in Figure 2.

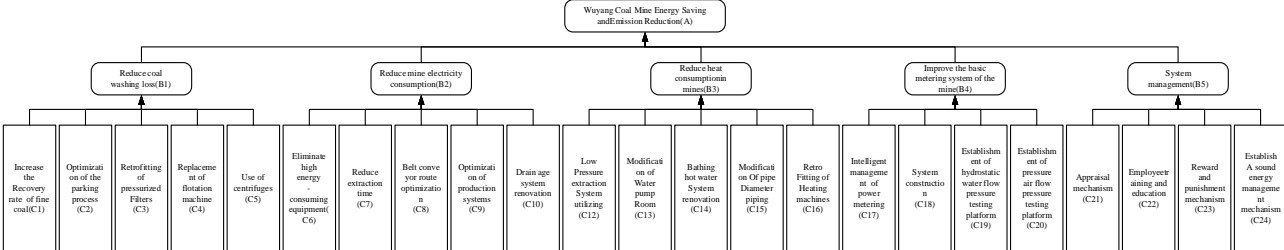

**Figure 2.** Hierarchical model of energy-saving and emission reduction in Wuyang coal mine.

### 3.3. Calculation of the Weight of Each Factor

3.3.1. The Relative Weight of the Criterion Layer to the Target Layer

According to the enterprise's energy consumption information provided by Wuyang coal mine and the project-related research and the actual situation in the mine, the degree of importance of each index is assigned by using the 1–9 comparison scale, which is assigned to each expert group to provide reference to rate the experts. Experts can score the elements

of each layer in the hierarchical structure according to the actual situation and refer to the scale in Table 1 to obtain the ratio between the two factors. If there is a large difference between the ratio of experts' scoring and the initial importance degree assignment, the relevant information will be consulted again and the site will be re-investigated, and the experts will be communicated with until the two assignments are approximately consistent.

The obtained importance results are arranged and the results are then written in the form of a matrix, which is used as the broken matrix of the judgment index, as shown in Table 3. Finally, the weight of each factor is and the maximum eigenvalue are calculated by using the square root method.

**Table 3.** A-B judgment matrix.

| Target Layer A | $B_1$ | $B_2$ | $B_3$ | $B_4$ | $B_5$ |
|---|---|---|---|---|---|
| $B_1$ | 1 | 2 | 2 | 3 | 4 |
| $B_2$ | 1/2 | 1 | 1 | 3/2 | 2 |
| $B_3$ | 1/2 | 1 | 1 | 3/2 | 2 |
| $B_4$ | 1/3 | 1 | 2/3 | 1 | 1 |
| $B_5$ | 1/4 | 2/3 | 2/3 | 1 | 1 |

According to Equation (1), we can find the weight vector of each factor $B_1$–$B_5$ $w'_i$:

$$w'_1 = 2.1689, w'_2 = 1.0844, w'_3 = 1.0844, w'_4 = 0.7402, w'_5 = 0.6443$$

The $w'_i$ is normalized and the evaluation factor weight vector $w_i$ is derived according to Equation (2):

$$w_1 = 0.3873, w_2 = 0.1936, w_3 = 0.1936, w_4 = 0.1224, w_5 = 0.1031$$
$$A = (0.3873, 0.1936, 0.1936, 0.1224, 0.1031)$$

According to Equation (3), the maximum eigenvalue can be found as $\lambda_{max} = 5.0100$. Checking Table 2 yields the fifth-order matrix of $R \cdot I = 1.12$, and substituting this into Equations (4) and (5) results in:

$$C \times R = \frac{\lambda_{max} - n}{(n-1)R \times I} = 0.0022 < 0.1$$

This then verifies that the matrix satisfies the consistency test, and that the requested eigenvectors are valid.

### 3.3.2. Calculation of the Relative Weights of the Alternative Layer to the Criterion Layer

(1)  Establish the judgment matrix of $B_1$

The factors affecting $B_1$ are scored according to the information provided in the literature [18] and the expert opinions collected on site from the Wuyang coal mine. The judgment matrix of $B_1$, the weight vector approximation $w'_i$, and the weight $w_i$ of each factor are derived, as shown in Table 4.

$$\lambda_{max} = 5.0198, W_{B1} = (0.3064, 0.1407, 0.2814, 0.1407, 0.1307)$$

**Table 4.** Judgment matrix of $B_1$–C.

| $B_1$ | $C_1$ | $C_2$ | $C_3$ | $C_4$ | $C_5$ | $w'_i$ | $w_i$ |
|---|---|---|---|---|---|---|---|
| $C_1$ | 1 | 2 | 1 | 2 | 3 | 1.6437 | 0.3064 |
| $C_2$ | 1/2 | 1 | 1/2 | 1 | 1 | 0.7578 | 0.1407 |
| $C_3$ | 1 | 2 | 1 | 2 | 2 | 1.5157 | 0.2814 |
| $C_4$ | 1/2 | 1 | 1/2 | 1 | 1 | 0.7578 | 0.1407 |
| $C_5$ | 1/3 | 1 | 1/2 | 1 | 1 | 0.6988 | 0.1307 |

By checking Table 2, we can determine that R·I = 1.12 of the fifth-order matrix, and substituting this into Equations (4) and (5) results in:

$$C \times R = \frac{\lambda_{max} - n}{(n-1)R \times I} = 0.0044 < 0.1$$

This means that the matrix satisfies the test of consistency, i.e., the requested eigenvectors are valid.

(2)   Judgment matrix for reducing mine electricity consumption $B_2$

The factors affecting $B_2$ are scored according to the information provided in the literature [19] and the expert opinions collected on site from the Wuyang coal mine. The judgment matrix of $B_2$, the weight vector approximation $w'_i$, and the weight $w_i$ of each factor are derived, as shown in Table 5.

$$\lambda_{max} = 6.0550, W_{B2} = (0.3171, 0.2379, 0.1586, 0.1586, 0.0916, 0.0362)$$

**Table 5.** Judgment matrix of $B_2$–C.

| $B_2$ | $C_6$ | $C_7$ | $C_8$ | $C_9$ | $C_{10}$ | $C_{11}$ | $w'_i$ | $w_i$ |
|---|---|---|---|---|---|---|---|---|
| $C_6$ | 1 | 4/3 | 2 | 2 | 4 | 8 | 2.3551 | 0.3171 |
| $C_7$ | 3/4 | 1 | 3/2 | 3/2 | 3 | 6 | 1.7663 | 0.2379 |
| $C_8$ | 1/2 | 2/3 | 1 | 1 | 2 | 4 | 1.1775 | 0.1586 |
| $C_9$ | 1/2 | 2/3 | 1 | 1 | 2 | 4 | 1.1775 | 0.1586 |
| $C_{10}$ | 1/4 | 1/3 | 1/2 | 1/2 | 1 | 4 | 0.6609 | 0.0916 |
| $C_{11}$ | 1/8 | 1/6 | 1/4 | 1/4 | 1/4 | 1 | 0.2622 | 0.0362 |

By checking Table 2, we can determine that R·I = 1.26 of the sixth-order matrix, and substituting this into Equations (4) and (5) results in:

$$C \times R = \frac{\lambda_{max} - n}{(n-1)R \times I} = 0.0087 < 0.1$$

This means that the matrix satisfies the test of consistency, i.e., the requested eigenvectors are valid.

(3)   Judgment matrix for reducing the heat consumption of the mine $B_3$

The factors affecting $B_3$ are scored according to the information provided in the literature [20] and the expert opinions collected on site from the Wuyang coal mine. The judgment matrix of $B_3$, the weight vector approximation $w'_i$, and the weight $w_i$ of each factor are derived, as shown in Table 6.

$$\lambda_{max} = 5.0297, W_{B3} = (0.2710, 0.2052, 0.1472, 0.2052, 0.1713)$$

**Table 6.** Judgment matrix of $B_3$–C.

| $B_3$ | $C_{12}$ | $C_{13}$ | $C_{14}$ | $C_{15}$ | $C_{16}$ | $w'_i$ | $w_i$ |
|---|---|---|---|---|---|---|---|
| $C_{12}$ | 1 | 5/4 | 2 | 5/4 | 5/3 | 1.3910 | 0.2710 |
| $C_{13}$ | 4/5 | 1 | 6/5 | 1 | 4/3 | 1.0506 | 0.2052 |
| $C_{14}$ | 1/2 | 5/6 | 1 | 5/6 | 2/3 | 0.7462 | 0.1472 |
| $C_{15}$ | 4/5 | 1 | 6/5 | 1 | 4/3 | 1.0506 | 0.2052 |
| $C_{16}$ | 3/5 | 3/4 | 3/2 | 3/4 | 1 | 0.8727 | 0.1713 |

By checking Table 2, we can determine that R·I = 1.12 of the fifth-order matrix, and substituting this into Equations (4) and (5) result in:

$$C \times R = \frac{\lambda_{max} - n}{(n-1)R \times I} = 0.0066 < 0.1$$

This means that the matrix satisfies the test of consistency, i.e., the requested eigenvectors are valid.

(4) Improve the judgment matrix of the mine base metering system $B_4$

The factors affecting $B_4$ are scored according to the information provided in the literature [21] and the expert opinions collected on site from the Wuyang coal mine. The judgment matrix of $B_4$, the weight vector approximation $w'_i$, and the weight $w_i$ of each factor are derived, as shown in Table 7.

$$\lambda_{max} = 4.0017, W_{B4} = (0.4624, 0.2246, 0.1588, 0.1541)$$

**Table 7.** Judgment matrix of $B_4$–C.

| $B_4$ | $C_{17}$ | $C_{18}$ | $C_{19}$ | $C_{20}$ | $w'_i$ | $w_i$ |
|-------|----------|----------|----------|----------|--------|-------|
| $C_{17}$ | 1 | 2 | 3 | 3 | 2.0597 | 0.4624 |
| $C_{18}$ | 1/2 | 1 | 4/3 | 3/2 | 1.0000 | 0.2246 |
| $C_{19}$ | 1/3 | 3/4 | 1 | 1 | 0.7071 | 0.1588 |
| $C_{20}$ | 1/3 | 2/3 | 1 | 1 | 0.6865 | 0.1541 |

By checking Table 2, we can determine that $R \cdot I = 0.89$ for the fourth-order matrix, and substituting into Equations (4) and (5) results in:

$$C \times R = \frac{\lambda_{max} - n}{(n-1)R \times I} = 0.0006 < 0.1$$

This means that the matrix satisfies the test of consistency, i.e., the requested eigenvectors are valid.

(5) Judgment matrix of system management $B_5$

The factors affecting $B_5$ are scored according to the information provided in the literature [22] and the expert opinions collected on site from the Wuyang coal mine. The judgment matrix of $B_5$, the weight vector approximation $w'_i$, and the weight $w_i$ of each factor are derived, as shown in Table 8.

$$\lambda_{max} = 4.0206, W_{B5} = (0.1718, 0.1718, 0.1910, 0.4654)$$

**Table 8.** Judgment matrix of $B_5$–C.

| $B_5$ | $C_{21}$ | $C_{22}$ | $C_{23}$ | $C_{24}$ | $w'_i$ | $w_i$ |
|-------|----------|----------|----------|----------|--------|-------|
| $C_{21}$ | 1 | 1 | 1 | 1/3 | 0.7598 | 0.1718 |
| $C_{22}$ | 1 | 1 | 1 | 1/3 | 0.7598 | 1.1718 |
| $C_{23}$ | 1 | 1 | 1 | 1/2 | 0.8408 | 0.1910 |
| $C_{24}$ | 3 | 3 | 2 | 1 | 2.5097 | 0.4654 |

By checking Table 2, we can determine that $R \cdot I = 0.89$ for the fourth-order matrix, and substituting this into Equations (4) and (5) results in:

$$C \times R = \frac{\lambda_{max} - n}{(n-1)R \times I} = 0.0077 < 0.1$$

This means that the matrix satisfies the test of consistency, i.e., the requested eigenvectors are valid.

*3.4. Factor Ranking*

The weight of reducing coal washing losses to energy saving and emission reduction in mines is:

$$W_1 = (0.1187, 0.0545, 0.1090, 0.0545, 0.0506)$$

The weight of reducing mine electricity consumption on mine energy saving and emission reduction is:

$$W_2 = (0.0614, 0.0461, 0.0307, 0.0307, 0.0177, 0.0070)$$

The weight of improving the basic metering system of mines on energy saving and emission reduction in mines is:

$$W_3 = (0.0525, 0.0397, 0.0285, 0.0397, 0.0332)$$

The weight of improving the basic measurement system of the mine on energy saving and emission reduction in mines is:

$$W_4 = (0.0566, 0.0275, 0.0194, 0.0189$$

The weight of institutional management on energy saving and emission reduction in mines is:

$$W_5 = (0.0177, 0.0177, 0.0197, 0.0480)$$

In summary, the order of importance of the factors related to energy saving and emission reduction in mines was discharged, as shown in Table 9.

**Table 9.** Weighting of the factors of energy saving and emission reduction in mines.

| Target Layer | Guideline Layer | Indicator Layer | | Sorting |
|---|---|---|---|---|
| | | Layering | Total Weighting | |
| Energy-saving and emission reduction in mines | B$_1$ (0.3873) | C$_1$ | 0.1187 | 1 |
| | | C$_2$ | 0.0545 | 4 |
| | | C$_3$ | 0.1090 | 2 |
| | | C$_4$ | 0.0545 | 4 |
| | | C$_5$ | 0.0506 | 7 |
| | B$_2$ (0.1936) | C$_6$ | 0.0614 | 3 |
| | | C$_7$ | 0.0461 | 9 |
| | | C$_8$ | 0.0307 | 13 |
| | | C$_9$ | 0.0307 | 13 |
| | | C$_{10}$ | 0.0177 | 20 |
| | | C$_{11}$ | 0.0070 | 24 |
| | B$_3$ (0.1936) | C$_{12}$ | 0.0525 | 6 |
| | | C$_{13}$ | 0.0397 | 10 |
| | | C$_{14}$ | 0.0285 | 15 |
| | | C$_{15}$ | 0.0397 | 10 |
| | | C$_{16}$ | 0.0332 | 12 |
| | B$_4$ (0.1224) | C$_{17}$ | 0.0566 | 4 |
| | | C$_{18}$ | 0.0194 | 18 |
| | | C$_{19}$ | 0.0275 | 16 |
| | | C$_{20}$ | 0.0189 | 19 |
| | B$_5$ (0.1031) | C$_{21}$ | 0.0177 | 20 |
| | | C$_{22}$ | 0.0177 | 20 |
| | | C$_{23}$ | 0.0197 | 17 |
| | | C$_{24}$ | 0.0480 | 8 |

Table 9 shows the importance of each factor affecting energy saving and emission reduction in Wuyang coal mine. According to the comparison of the weight values, the main measure of energy saving and emission reduction in Wuyang coal mine is to improve the recovery rate of clean coal. Because the coal mine is a non-renewable resource, improving the recovery rate of clean coal can effectively improve the utilization rate of resources and reduce unnecessary waste. The next step is to modify the pressurized filters and eliminate high energy-consuming equipment. This measure is easy to operate and will reduce the moisture of the flotation concentrate; eliminating high energy-consuming equipment can reduce the energy loss from the root cause, thus achieving the effect of energy saving and emission reduction. In addition to the above points, the optimization of the stopping

process, replacement of flotation machines, utilization of low pressure extraction system, using centrifuges, establishment of a sound energy management mechanism and reduction in extraction time all play a dominant role in the overall energy-saving and emission reduction project. Although institutional management accounts for a relatively small proportion of influencing factors, it is still indispensable, and the existence of a suitable and perfect system support means that energy-saving and emission reduction projects can be carried out efficiently.

### 3.5. Fuzzy Integrated Evaluation Method

3.5.1. The Use of Expert Scoring to Obtain a Set of Qualitative Comments

Experts are invited to score the energy-saving and emission reduction in Wuyang coal mine according to the index evaluation grade standard. The grade description of the relevant indicators in the references shows that six indicators are taken out for each influencing factor, and five grades of "excellent, good, medium, poor, and bad " are established. The grade standard is shown in Table 10. Then, the percentage system method is used to count the scores of experts, and finally, the qualitative index comment set can be obtained, as shown in Table 11.

**Table 10.** Grade standard evaluation table.

| Indicator Name | Excellent (95 Points) | Good (80 Points) | Medium (65 Points) | Poor (50 Points) | Bad (30 Points) |
|---|---|---|---|---|---|
| $C_1$ | Complete clean coal recovery system with high recovery rate | Complete clean coal recovery system with average recovery rate | The clean coal recovery system is relatively completely and the recovery rate is average | The clean coal recovery system is still incomplete and the recovery efficiency is not high | The clean coal recovery system is very incomplete and has very low recovery efficiency |
| $C_6$ | The equipment and processes used are more advanced and do not consume energy | The equipment and process used are more advanced and slightly energy-consuming | The equipment and processes used are not advanced, and the energy consumption is average | The equipment and technology used are backward, slightly high energy-consuming | The equipment and technology used are very backward and very energy-consuming |
| $C_9$ | The production system is very well developed | More complete production system | General perfection of production system | Imperfect production system | Very imperfect production system |
| $C_{12}$ | Low pressure extraction systems are widely utilized and fully operational | Low-pressure extraction systems are more widely utilized and not fully operational | Low-pressure extraction systems are less widely utilized and operate well | Low-pressure extraction systems are not widely utilized and are in general operation | Low-pressure extraction systems are not widely utilized and not operated |
| $C_{17}$ | Sound information management system and very intelligent management | Information management system is relatively sound and intelligently managed | Basic sound information management system and intelligent management | Information management system is not very sound and slightly intelligent management | Information management system is very unsound and not intelligently managed |
| $C_{24}$ | The system is totally scientific, perfect, and well implemented | The system is more scientific and perfect, and the implementation is good | The system is generally scientific and sound, and the implementation is also average | The system lacks scientificity and perfection, and is poorly implemented | The system lacks science, integrity, and is very poorly implemented |

For example, ten experts were invited to evaluate the grade of "$C_1$ improve the recovery rate of refined coal", and two of them thought the grade was "excellent", so the total number of experts could be removed to obtain "excellent". The subordinate value of "excellent" is "0.2"; three persons were rated as "good" and four persons were rated as "medium", and the membership values of "good" and "medium" were "0.3" and "0.4", respectively. One person felt that the rating was " poor" and no one thought the rating was "bad", so, the values of "poor" and "bad" were "0.1" "0", respectively. In summary, the fuzzy evaluation matrix of $C_1$ is {0.2, 0.3, 0.4, 0.1, 0}.

**Table 11.** Summary of qualitative index score table.

| Guideline Layer B | Alternative Solution Layer C | Excellent | Good | Medium | Poor | Bad |
|---|---|---|---|---|---|---|
| B₁ | $C_1$ | 0.2 | 0.3 | 0.4 | 0.1 | 0 |
|  | $C_2$ | 0 | 0.1 | 0.5 | 0.3 | 0.1 |
|  | $C_3$ | 0.1 | 0.1 | 0.4 | 0.3 | 0.1 |
|  | $C_4$ | 0.1 | 0.3 | 0.2 | 0.4 | 0 |
|  | $C_5$ | 0.2 | 0.2 | 0.4 | 0.1 | 0.1 |
| B₂ | $C_6$ | 0.2 | 0.3 | 0.3 | 0.1 | 0.1 |
|  | $C_7$ | 0.1 | 0.5 | 0.2 | 0.2 | 0 |
|  | $C_8$ | 0 | 0.4 | 0.3 | 0.2 | 0.1 |
|  | $C_9$ | 0.3 | 0.2 | 0.3 | 0.1 | 0.1 |
|  | $C_{10}$ | 0 | 0.2 | 0.5 | 0.3 | 0 |
|  | $C_{11}$ | 0 | 0.3 | 0.4 | 0.2 | 0.1 |
| B₃ | $C_{12}$ | 0.2 | 0.2 | 0.2 | 0.2 | 0.2 |
|  | $C_{13}$ | 0.4 | 0.1 | 0.2 | 0.3 | 0 |
|  | $C_{14}$ | 0.2 | 0.2 | 0.4 | 0.2 | 0 |
|  | $C_{15}$ | 0.1 | 0.2 | 0.6 | 0 | 0.1 |
|  | $C_{16}$ | 0.2 | 0.4 | 0.3 | 0.1 | 0 |
| B₄ | $C_{17}$ | 0 | 0.3 | 0.5 | 0.1 | 0.1 |
|  | $C_{18}$ | 0.1 | 0.4 | 0.4 | 0 | 0.1 |
|  | $C_{19}$ | 0.3 | 0.2 | 0.1 | 0.4 | 0 |
|  | $C_{20}$ | 0.3 | 0.1 | 0.2 | 0.2 | 0.2 |
| B₅ | $C_{21}$ | 0.4 | 0.3 | 0.2 | 0 | 0.1 |
|  | $C_{22}$ | 0.2 | 0.3 | 0.3 | 0.3 | 0 |
|  | $C_{23}$ | 0.1 | 0.4 | 0.4 | 0.1 | 0 |
|  | $C_{24}$ | 0 | 0.2 | 0.4 | 0.3 | 0.1 |

3.5.2. One-Level Comprehensive Evaluation

(1) The fuzzy evaluation matrix of the alternative level can be obtained according to the qualitative index scoring table Ri

$$R_1 = \begin{bmatrix} 0.2 & 0.3 & 0.4 & 0.1 & 0 \\ 0 & 0.1 & 0.5 & 0.3 & 0.1 \\ 0.1 & 0.1 & 0.4 & 0.3 & 0.1 \\ 0.1 & 0.3 & 0.2 & 0.4 & 0 \\ 0.2 & 0.2 & 0.4 & 0.1 & 0.1 \end{bmatrix}$$

$$R_2 = \begin{bmatrix} 0.2 & 0.3 & 0.3 & 0.1 & 0.1 \\ 0.1 & 0.5 & 0.2 & 0.2 & 0 \\ 0 & 0.4 & 0.3 & 0.2 & 0.1 \\ 0.3 & 0.2 & 0.3 & 0.1 & 0.1 \\ 0 & 0.2 & 0.5 & 0.3 & 0 \\ 0 & 0.3 & 0.4 & 0.2 & 0.1 \end{bmatrix}$$

$$R_3 = \begin{bmatrix} 0.2 & 0.2 & 0.2 & 0.2 & 0.2 \\ 0.4 & 0.1 & 0.2 & 0.3 & 0 \\ 0.2 & 0.2 & 0.4 & 0.2 & 0 \\ 0.1 & 0.2 & 0.6 & 0 & 0.1 \\ 0.2 & 0.4 & 0.3 & 0.1 & 0 \end{bmatrix}$$

$$R_4 = \begin{bmatrix} 0 & 0.3 & 0.5 & 0.1 & 0.1 \\ 0.1 & 0.4 & 0.4 & 0 & 0.1 \\ 0.3 & 0.2 & 0.1 & 0.4 & 0 \\ 0.3 & 0.1 & 0.2 & 0.2 & 0.2 \end{bmatrix}$$

$$R_5 = \begin{bmatrix} 0.4 & 0.3 & 0.2 & 0 & 0.1 \\ 0.2 & 0.3 & 0.3 & 0.3 & 0 \\ 0.1 & 0.4 & 0.4 & 0.1 & 0 \\ 0 & 0.2 & 0.4 & 0.3 & 0.1 \end{bmatrix}$$

(2) Find the first-order integrated parity matrix Bi

According to the Equation (6), the first-level comprehensive evaluation matrix can be obtained as follows:

$$B_1 = W_{B1} \times R_1 = (0.1296\ 0.2025\ 0.3859\ 0.2267\ 0.0553)$$

Similarly available:

$$B_2 = W_{B2} \times R_2 = (0.1348\ 0.3384\ 0.2981\ 0.1616\ 0.0671)$$
$$B_3 = W_{B3} \times R_3 = (0.2205\ 0.2137\ 0.3286\ 0.1623\ 0.0747)$$
$$B_4 = W_{B4} \times R_4 = (0.1163\ 0.2757\ 0.3677\ 0.1406\ 0.0996)$$
$$B_5 = W_{B5} \times R_5 = (0.1222\ 0.2726\ 0.3485\ 0.2103\ 0.0637)$$

Then, the total evaluation matrix R is:

$$R = \begin{bmatrix} 0.1296 & 0.2025 & 0.3859 & 0.2267 & 0.0553 \\ 0.1348 & 0.3384 & 0.2981 & 0.1616 & 0.0671 \\ 0.2205 & 0.2137 & 0.3286 & 0.1623 & 0.0747 \\ 0.1163 & 0.2757 & 0.3677 & 0.1406 & 0.0996 \\ 0.1222 & 0.2726 & 0.3485 & 0.2103 & 0.0637 \end{bmatrix}$$

(3)　Secondary comprehensive evaluation

By the formula

$$B = A \times R \tag{8}$$

The following can be obtained:

$$B = A \times R = (0.15\ 0.25\ 0.35\ 0.19\ 0.06)$$

(4)　Grade evaluation

According to Equation (7), the following can be obtained:

$$f_1 = 95 \times 0.1296 + 80 \times 0.2025 + 65 \times 0.3859 + 50 \times 0.2267 + 30 \times 0.0553 = 66$$
$$f_2 = 95 \times 0.1348 + 80 \times 0.3384 + 65 \times 0.2981 + 50 \times 0.1616 + 30 \times 0.0671 = 69$$
$$f_3 = 95 \times 0.2205 + 80 \times 0.2137 + 65 \times 0.3286 + 50 \times 0.1623 + 30 \times 0.0747 = 69$$
$$f_4 = 95 \times 0.1163 + 80 \times 0.2757 + 65 \times 0.3677 + 50 \times 0.1406 + 30 \times 0.0996 = 67$$
$$f_5 = 95 \times 0.1222 + 80 \times 0.2726 + 65 \times 0.3485 + 50 \times 0.2103 + 30 \times 0.0637 = 68$$
$$f_{all} = 95 \times 0.15 + 80 \times 0.25 + 65 \times 0.35 + 50 \times 0.19 + 30 \times 0.06 = 68$$

According to the calculation results of the above scores, the experts' evaluation grades of the five influencing factors are "good". According to the total score of the system, the score level of the whole coal mine is also "good".

(5)　Evaluation results

The potential of energy saving and emission reduction in Wuyang coal mine was evaluated using a combination of hierarchical analysis and fuzzy comprehensive evaluation method. The evaluation results show that the rating level of the whole coal mine is good, the energy-saving and emission reduction potential of Wuyang coal mine is large, and there is still significant room for improvement [23,24]. According to the first level of weighting index values, it can be seen that the weighting value of reducing coal washing losses (0.3873) is the largest, and the weighting values of reducing mine electricity consumption (0.1936) and reducing mine heat consumption (0.1936) are equal, followed by improving the mine's basic measurement system (0.1224) and system management (0.1031). Among the indicators for reducing coal washing losses, the ones with higher weight values are increasing the recovery rate of clean coal (0.1187) and renovating the pressurized filter (0.1090). Among the indicators for reducing mine electricity consumption, the one with a higher weight is to eliminate high energy-consuming equipment (0.0614). Among the

indicators for reducing heat consumption in mines, the one with the highest weight is the utilization of low-pressure extraction system (0.0525). Among the indicators for improving the basic metering system of the mine, the one with the highest weight is the intelligent management of power metering (0.0566). Among the indicators for system management, the one with the greatest weight is the establishment of a sound energy consumption management system (0.0480). Among these indicators, the one with the greatest weight is to improve the recovery rate of clean coal. Because coal resources are non-renewable, to reduce energy consumption, we need to make measures on energy utilization rate, if we improve the recovery rate of clean coal, we will greatly reduce the loss of energy.

**4. Suggestions for Energy Saving and Emission Reduction in Mines**

From the above comparison of the weight of energy-saving and emission reduction factors in Wuyang coal mine, it can be seen that improving the recovery rate of refined coal (0.1187), renovating the pressure filter (0.1090), eliminating high energy-consuming equipment (0.0614), enacting an intelligent management of power metering (0.0614), and establishing a sound energy consumption management mechanism (0.0480) are the main factors in the energy-saving and emission reduction work of the mine. Combined with the actual situation of Wuyang coal mine, the corresponding countermeasures and measures can be proposed for the energy saving and emission reduction in the mine from the following aspects.

(1) Improve the recovery rate of clean coal

The coal recovery work of coal enterprises involves the optimization and change of all aspects of coal development and utilization. Even if the coal recovery work needs to pay a lot of manpower, material, and financial resources in the early stage, once the coal recovery work is successfully completed, the whole coal enterprise will bring great promotion to its own development. In this regard, we can adjust the sorting parameters of the heavy medium system, appropriately increase the ash content of the refined coal product, and improve the quality of the refined coal count on the premise of ensuring the quality index of the refined coal product. According to the on-site research of Wuyang coal mine and the consultation of professionals, it can be seen that the inadequate circulating water system of the mine is one of the main reasons for the low recovery rate of the coal mine. By improving the circulating water system, clean water can be ensured to wash coal, and the monitoring frequency of the clear water layer of the slimewater in the thickening pool is increased. At the same time, a desmear equipment can be added to the circulating water pipeline to reduce the blockage of the nozzle as much as possible and change the circulating water quality and strengthen the screening effect and improve the efficiency of medium removal. Adjusting the sieves plate and water spray setting of the dilute medium section of the desintering screen and adding a water spray at the desintering screen of clean coal can effectively improve the desintering efficiency and production efficiency and avoid the coal washing being affected by low density. Improve equipment transport capacity and safety performance. The maintenance of equipment should be strengthened to prevent the impact of equipment damage on working hours and efficiency.

(2) Equipment replacement and transformation

Equipment renovation is the guarantee of expanding reproduction and saving energy. With the rapid development of modern industrial society, the mining work of coal industry has become more and more demanding in terms of enterprise technology and equipment. Although Wuyang coal mine has updated a part of its machinery and equipment, the flotation machine is still selected from the old equipment, and the single flotation effect is poor, which reduces the energy usage rate to a certain extent. If the new clean coal centrifuge is used, it may reduce the moisture of heavy-duty clean coal, thus increasing the heat of heavy-duty clean coal, which will improve the utilization rate of coal mine resources. Although the replacement equipment can achieve results quickly, in practice, it is difficult for many companies to meet all the requirements of updating equipment with available

funds, and on this basis, retrofitting old equipment is a very effective way. On the basis of the old equipment to locally update, the production adaptability of the old equipment is enhanced, and sometimes will achieve the effect of less investment and quicker results. For example, remodeling the pressurized filter, where the collected filtrate water of the pressurized filter is poor, and is then transferred to the slurry pre-processor of the flotation system by the slurry pump, forming a small closed-circuit cycle to recover all the low-ash fine-grained coal slurry to the clean coal. Furthermore, some energy-consuming equipment constantly consume limited resources, which cause certain losses to the environment and the economic benefits of the enterprise. Outdated high energy-consuming equipment and processes can be eliminated or upgraded to achieve the purpose of reducing energy consumption.

(3)     Intelligent management of power metering

Power metering is a very important part of energy saving and consumption reduction, and the intelligent management of power metering can improve and optimize the rules and regulations, and the focus of its work is to check the metering instruments. Through the field investigation of Wuyang coal mine, the power consumption of the raw coal production process is systematically analyzed, and some electronic monitoring instruments with low precision are found. If these instruments are updated, the resource loss caused by inaccurate equipment can be avoided as much as possible. It can also carry out the pilot project of electricity efficiency control, select the representative mining face, and reasonably quantify and manage the electricity consumption of main electricity equipment according to the actual electricity consumption combined with raw coal output and excavation extension meters.

(4)     System management

In addition to reducing energy consumption by improving equipment, we should also strengthen the technical training of management personnel, carry out training for energy-saving technicians, deepen the technicians' understanding of energy saving and emission reduction, and improve their working ability. Moreover, building a system of energy-saving technical personnel and increasing the proportion of professional staff are also significant. The mine should also establish an assessment mechanism to assess the completion of energy consumption work targets on a monthly basis, and units that exceed monthly consumption will receive treatment. Each energy-using unit in charge of the mine can improve the management of the unit according to the "Standardized Management System Construction Standard for Double Control of Energy Consumption of Lu'an Chemical Group", and if the incomplete items are found, the relevant responsibilities will be pursued. An incentive mechanism can also be established. In the work of the double control of energy consumption, relevant units that complete the measures on schedule and the monthly indexes can be rewarded with cash. The assessment results of double control of energy consumption will be used as an important basis for the selection of other awards in the year, so as to mobilize the enthusiasm of employees.

## 5. Conclusions

This paper describes the specific application method of fuzzy hierarchical analysis in the assessment of energy-saving and emission reduction potential of mines by taking Wuyang coal mine in Shanxi Province of China as an example. Based on the project evaluation set, a reasonable evaluation system is constructed to carry out the evaluation, and then fuzzy arithmetic is used to obtain the assessment results. Since the project evaluation set is determined by the project program and the actual construction of the project, the method is universal. According to the evaluation results and the deficiencies in energy-saving and emission reduction work of Wuyang coal mine, the corresponding countermeasures are proposed. The conclusions are as follows:

(1)     According to the documents and information about energy saving and emission reduction in Wuyang coal mine, and the discussion of experts, the factors affecting

energy saving and emission reduction can be divided into two levels. It includes five criteria layers and 24 alternative layers. According to the data, the criteria layer can be divided into reducing coal washing loss, reducing mine power consumption, reducing mine heat consumption, and improving mine basic measurement system and system management. The largest weight is to reduce coal washing loss (0.3873), reducing mine electricity consumption (0.1936), and reducing mine heat consumption (0.1936), which have equal weight, followed by improving mine basic metering system (0.1224), and the smallest weight is the system management (0.1031).

(2) According to the weight ranking, the main factors affecting energy saving and emission reduction can be obtained in 24 alternative schemes, such as improving the recovery rate of clean coal, reforming the pressure filter, eliminating the high energy-consuming equipment, optimizing the parking process, replacing the flotation machine, and the intelligent management of power metering. The greatest weight is to improve the recovery of clean coal (0.1187), followed by modifying the pressurized filter (0.1090) and elimination of high energy-consuming equipment (0.0614).

(3) The fuzzy comprehensive evaluation of the weight obtained by the analytic hierarchy process can obtain the energy-saving and emission reduction situation of Wuyang coal mine. The evaluation grade of the five factors of the energy-saving and emission reduction scheme layer of Wuyang coal mine is 'good'. It can be seen that Wuyang coal mine has great potential for energy conservation and emission reduction, and there is a lot of room for improvement. Corresponding countermeasures can be proposed for the factors with higher weight in Wuyang coal mine, such as improving the recovery rate of refined coal, upgrading and modernizing energy-consuming and old equipment, perfecting the basic measurement system the mine, and establishing a sound energy consumption management mechanism.

**Author Contributions:** Writing—original draft preparation and collecting field data, F.X.; writing—review and editing and formal analysis, R.H.; provision of experimental protocols and geometric modeling, M.Z.; major translation work, W.Z.; data processing and drawing, Q.K.; modification of article format, M.D. All authors have read and agreed to the published version of the manuscript.

**Funding:** This research was funded by the National Natural Science Foundation of China (52174086), the National Natural Science Foundation of China Youth Fund (51804222), the 14th Graduate Innovative Fund of Wuhan Institute of Technology (CX2022577), the 2022 Hubei Master Teacher Studio, and the Educational Commission of Hubei Province of China (D20201506).

**Institutional Review Board Statement:** Not applicable.

**Informed Consent Statement:** Not applicable.

**Data Availability Statement:** Not applicable.

**Conflicts of Interest:** The authors declare no conflict of interest.

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
