# Peer review of "Energy Saving and Emission Reduction Potential Evaluation of a Coal Mine Based on Fuzzy Hierarchical Analysis"

_sustainability, doi:10.3390/su151511754_

Round 1
Reviewer 1 Report (Previous Reviewer 4)
This work is actually a resubmitted version of a previously reviewed manuscript (sustainability-2415577). The current version shows improvement with respect to the previous one; however, additional action still appears strongly advised. In general, a more quantitative insight is recommended, rather than engaging in lengthy rebuttals to reviewers or adding lengthy parts to the text. The comments may refer to the authors’ response.
Point 1. You may clearly state how your methodology and results may be of general applicability, both in the introduction and in the conclusions. I see those sections have been expanded with a large amount of text, but they still somewhat fail to respond to that question.
Point 2. “Based on the enterprise energy consumption… …the judgment matrix is listed as shown in Table 3.”: unfortunately, that does not explain how you assessed the two-by-two comparison. You may at least describe the whole process, from actual situation through expert opinion, for a couple of instances.
Point 3. Why are those five and 24 entries listed in Fig. 2 enough to describe all the aspects involved in energy saving and emission reduction for a coal mine?
Point 5. The introduction has been made even lengthier, but the reason why you cite those sources is still unclear: how did they serve as a foundation for your work?
The current abstract is so long that it actually appears more similar to a regular section than an abstract.
The overall quality of the manuscript is still rather poor: inappropriate blank spaces (e.g., “have large deviations .”), inconsistent citations (e.g., “Witold,P et al.[3]”), section titles lost in the middle of the text (e.g., sections 2 and 5), unexplained acronyms (e.g., CI of Fig. 1).
You may see the comments and suggestions.
Author Response
Dear Experts,
I have carefully revised the paper as per your comments, please refer to the attached.

Reviewer 2 Report (Previous Reviewer 2)
This manuscript has been improved compared with the first version. However, this version is not good enough to publish. Please revise as per reviewer’s comments to improve to the higher quality.
Abstract: The abstract was improved compared with the previous version. But it still not well organized, some duplicate sentences were detected as well as “This paper takes Wuyang Coal Mine in Shanxi Province, China as an example”. In addition, total words in abstract should not over than 250 words respect to the “guild for author” of Sustainability.
Introduction: The introduction was improved but still not well organized in the literature part especially for text citation and many typos have been detected for example “Double full stop shows in many time”. Please kindly check.
Case Studies: Subsection 3.1: Establishment of the indicator system.
Please kindly check the subsection alignment (2) and (3), it must be place at the beginning of new paragraph.
Table 4-7; it is better for the reader if the authors can combine table 4-8 into one table including explanation of routine step.
Table 9; Due to a lot of text description in each variable, the table will lack attraction from the readers. So, the author can increase the quality of this table by showing the Variable letter rather than show it definition for example “Reduce coal washing losses” must be “B1”, “Increase the recovery rate of fine coal” must be “C1”.
Please carefully check typoes and format.
Author Response
Dear Experts,
I have carefully revised the paper as per your comments, please refer to the attached.

Round 2
Reviewer 1 Report (Previous Reviewer 4)
No further comments.
No further comments.
This manuscript is a resubmission of an earlier submission. The following is a list of the peer review reports and author responses from that submission.
Round 1
Reviewer 1 Report
See attached annotated manuscript.

Punctuations and referencing style are very poor throughout the manuscritp
Reviewer 2 Report
This manuscript investigates the energy conservation and emission reduction in Wuyang Coal Mine (China) by using Fuzzy Hierarchy analysis. Many factors that affect energy conservation and emission reduction have been evaluated and put forward to provide guidance for future work. However, this version is not good enough to publish. Please revise the manuscript to improve it to higher quality.
Abstract:
The abstract is poorly managed. It lacks research background and research objective that make the readers unclearly understand why this work was study. There are many questions that were generated while reading this abstract, what is Fuzzy hierarchy? Why Wuyang Coal Mine are devided into 24?
1. Introduction:
The overall content of the introduction was not well managed, especially for paragraph 2 which relates to the recent literature, this content should be reorganized. The authors must provide the information of fuzzy and hierarchy analysis and leading the reader to know about How it operates and how it’s beneficial on decision making. In addition, the background of Wuyang coal mine must have been presented as well. The authors should revise this section and provide the content follow the standard structure of introduction including background, related literatures, methodology, research gap, objective and expected outcome.
2 Analytic Hierarchy Process and fuzzy comprehensive evaluation:
It is better for the readers, if the author provides the infographic to represent the method of AHP and fuzzy.
There are many equations that were presented without mention in the main text. In addition, the authors should provide the definition of every variable presented in each equation.
3. Project examples:
Subsection 3.1: This section should be placed in the introduction part.
Subsection 3.2: Why this subsection was categorized into 5 categories, please provide a more detailed reference/explanation.
Subsection 3.3: Due to lack of background information and no evidence of data analysis presented from subsection 3.2, leading the readers to be confused about why the authors identify the factors as 24 factors.
Subsection 3.4: This section has no need to show the detail of calculation (The authors can move this detail into supplementary). In addition, the authors must show the significant meaning of weight factor and relationship of each factor.
Subsection 3.5–3.6: These two subsections are the principal part of this study. The authors should restructure the whole content of this manuscript to pay attention on this content rather than give a detail on calculation.
4. Wuyang coal mine energy saving measures:
This section was provided with poorly managed. I recommend that the authors should present the results from fuzzy and hierarchy analysis belong with the explanation on the operational activities of Wuyang Coal Mine. On the other hand, this section must be combined with section 3.
Reference:
All reference lists were organized with the wrong format. Please kindly revise it is following the “Guide for authors” found on Sustainability homepage.
Please kindly edit many mistakes as list below
- Figure 1: Some arrow lines overlapped. In addition, please identify the target layer, guideline layer and alternative layer in this figure.
- There are some abbreviations that were used without full spelling out on the first time of mention.
- Please check text citations for example “LAA RHOVEN PJM and PEDRYCZ Wi” should be revised following with author guide found in Sustainability homepage.
Reviewer 3 Report
This study provides new inspiration for the development of energy saving and emission reduction strategies for mines under the "double carbon target", which is conducive to solving the important problem of resource wastage in mines and continuously improving the economic efficiency of coal enterprises so as to achieve the long-term and stable development of coal enterprises. 1)Further elaboration on Wuyang Coal Mine's representativeness and research method generalizability; 2) Analyzing shortcomings of existing policies and future policy recommendations is suggested to add.
Minor editing of English language required.
Reviewer 4 Report
This work presents an analysis of some parameters of interest for improving energy consumption and reducing emissions in a coal mine through a fuzzy model. The objectives may fall within the scopes of Sustainability, yet the contents appear somewhat out of focus. Moreover, the original contribution is slightly unclear.
Main:
1. This work is merely a case study, with very little link to any general applicability, even to other coal mines. Fuzzy modeling can be used to analyze several scenarios in a variety of applications, so the employed model may be contextualized and possibly validated to assess energy saving and emission reduction to many purposes. Unfortunately, that is almost fully missing, thus leaving a very specific set of information as the only contribution.
2. The hierarchical model used in this work is quite classic, so it would have been interesting to know how the judgement matrices presented in Tables 3 – 8 were generated, as that is the starting point of the whole process. Unfortunately, the assessment of the two-by-two relative importance is almost fully missing, since the figures are merely shown in those tables, which in fact would have a been a contribution of yours in terms of evaluating how significant are the selected parameters with respect to each other in a mine scenario.
3. Very much related to the previous point, the two layers presented in Fig. 1 (B and C) may actually be reasonable, but they are completely unsubstantiated: why those five and those 24? What do they represent and do they represent all the instances to be considered to perform energy saving and reduce emissions in a coal mine? If so, why?
4. Section 4 would be expected as the actual contribution to research on sustainability, since the whole work is aimed at reaching energy saving and emission reduction. However, that part is merely a list of actions, some of which are included in the selected layers and some others are unknown (e.g., “The new clean coal centrifuge”). Even more importantly, some claimed results (e.g., “reduce the moisture content of heavy medium clean coal by 0.2 % and increase the calorific value of heavy medium clean coal”, “The target completion of energy consumption work is assessed on a monthly basis, and the units with monthly overconsumption will be processed.”) are stated without being substantiated at all. Moreover, no prioritization is introduced in Section 4, which should be the main outcome from the hierarchical model.
5. The introduction includes an extensive part (“Due to the complexity of mine production… … using the improved hierarchical analysis [10].”), where a survey of various literature sources is presented. The next paragraph starts with “In view of this”: in fact, there is no clear connection between the reviewed studies and yours. No answer to the following question: which of them was useful for your work and how?
Additional comments are listed below.
The five grades listed in Subsection 3.6 are “excellent, good, medium, poor and inferior”, while Table 10 reports “Excellent”, “Good”, “Medium”, “Poor” and “Bad”: consistency is required.
The use of “etc.” may be avoided in scientific communication: you may either simply remove it or remove it and list all the items.
Acronyms may be spelled out the first time they appear (e.g., MBNQA, OHS).
A revision of the whole manuscript is advised to correct a few grammar mistakes (e.g., “then” at the beginning of a sentence), typos (e.g., “When ,Then the…”, “B1, B2, B3,perfect B4 and B5”) and parts where hyphenation went missing (e.g., “the requested eigenvect ors”, “and em ission reduction”).
You may consider the general comments as a reference.